# SOLVING COMPOSITIONAL REINFORCEMENT LEARNING PROBLEMS VIA TASK REDUCTION

**Yunfei Li**[1,♯]**, Yilin Wu**[2]**, Huazhe Xu**[3]**, Xiaolong Wang**[4]**, Yi Wu**[1,2,♮]

[1] Institute for Interdisciplinary Information Sciences, Tsinghua University
[2] Shanghai Qi Zhi Institute, [3] UC Berkeley, [4] UCSD

♯`liyf20@mails.tsinghua.edu.cn`, ♮`jxwuyi@gmail.com`

## ABSTRACT

We propose a novel learning paradigm, _Self-Imitation via Reduction (SIR)_, for solving compositional reinforcement learning problems. SIR is based on two core ideas: _task reduction_ and _self-imitation_. Task reduction tackles a hard-to-solve task by actively reducing it to an easier task whose solution is known by the RL agent. Once the original hard task is successfully solved by task reduction, the agent naturally obtains a self-generated solution trajectory to imitate. By continuously collecting and imitating such demonstrations, the agent is able to progressively expand the solved subspace in the entire task space. Experiment results show that SIR can significantly accelerate and improve learning on a variety of challenging sparse-reward continuous-control problems with compositional structures. Code and videos are available at `https://sites.google.com/view/sir-compositional`.

## 1 INTRODUCTION

A large part of everyday human activities involves sequential tasks that have compositional structure, i.e., complex tasks which are built up in a systematic way from simpler tasks (Singh, 1991). Even in some very simple scenarios such as lifting an object, opening a door, sitting down, or driving a car, we usually decompose an activity into multiple steps and take multiple physical actions to finish each of the steps in our mind. Building an intelligent agent that can solve a wide range of compositional decision making problems as such remains a long-standing challenge in AI.

Deep reinforcement learning (RL) has recently shown promising capabilities for solving complex decision making problems. But most approaches for these compositional challenges either rely on carefully designed reward function (Warde-Farley et al., 2019; Yu et al., 2019; Li et al., 2020; OpenAI et al., 2019), which is typically subtle to derive and requires strong domain knowledge, or utilize a hierarchical policy structure (Kulkarni et al., 2016; Bacon et al., 2017; Nachum et al., 2018; Lynch et al., 2019; Bagaria & Konidaris, 2020), which typically assumes a set of low-level policies for skills and a high-level policy for choosing the next skill to use. Although the policy hierarchy introduces structural inductive bias into RL, effective low-level skills can be non-trivial to obtain and the bi-level policy structure often causes additional optimization difficulties in practice.

In this paper, we propose a novel RL paradigm, _Self-Imitation via Reduction (SIR)_, which (1) naturally works on sparse-reward problems with compositional structures, and (2) does not impose any structural requirement on policy representation. SIR has two critical components, _task reduction_ and _imitation learning_. Task reduction leverages the compositionality in a _parameterized task space_ and tackles an unsolved hard task by actively "_simplifying_" it to an easier one whose solution is known already. When a hard task is successfully accomplished via task reduction, we can further run _imitation learning_ on the obtained solution trajectories to significantly accelerate training.

Fig. 1 illustrates _task reduction_ in a pushing scenario with an elongated box (brown) and a cubic box (blue). Consider pushing the cubic box to the goal position (red). The difficulty comes with the wall on the table which contains a small door in the middle. When the door is clear (Fig. 1a), the task is easy and straightforward to solve. However, when the door is blocked by the elongated box (Fig. 1b), the task becomes significantly harder: the robot needs to clear the door before it can start pushing the

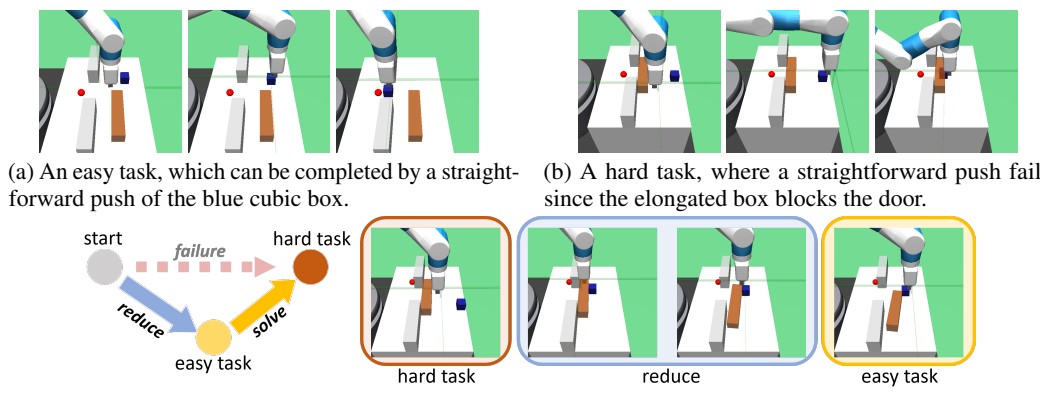

(a) An easy task, which can be completed by a straightforward push of the blue cubic box.

(b) A hard task, where a straightforward push fails since the elongated box blocks the door.

(c) *Task reduction*: solve a hard task by reducing it to an easier one.

Figure 1: An example use case of *task reduction*, the key technique of our approach. The task is to push the blue box to the red position.

cubic box. Such a compositional strategy is non-trivial to discover by directly running standard RL. By contrast, this solution is much easier to derive via task reduction: starting from an initial state with the door blocked (Fig. 1b), the agent can first imagine a simplified task that has the same goal but starts from a different initial state with a clear door (Fig. 1a); then the agent can convert to this simplified task by physically perturbing the elongate box to reach the desired initial state (Fig. 1c); finally, a straightforward push completes the solution to the original hard task (Fig. 1b).

Notably, by running imitation learning on the composite reduction trajectories, we effectively incorporate the inductive bias of compositionality into the learned policy without the need of explicitly specifying any low-level skills or options, which significantly simplifies policy optimization. Moreover, although task reduction only performs 1-step planning from a planning perspective, SIR still retains the capability of learning an *arbitrarily complex* policy by alternating between imitation and reduction: as more tasks are solved, these learned tasks can *recursively* further serve as new reduction targets for unsolved ones in the task space, while the policy gradually generalizes to increasingly harder tasks by imitating solution trajectories with growing complexity.

We implement SIR by jointly learning a goal-conditioned policy and a universal value function (Schaul et al., 2015), so that task reduction can be accomplished via state search over the value function and policy execution with different goals. Empirical results show that SIR can significantly accelerate and improve policy learning on challenging *sparse-reward* continuous-control problems, including robotics pushing, stacking and maze navigation, with both object-based and visual state space.

## 2 RELATED WORK

Hierarchical RL (HRL) (Barto & Mahadevan, 2003) is perhaps the most popular approach for solving compositional RL problems, which assumes a low-level policy for primitive skills and a high-level policy for sequentially proposing subgoals. Such a bi-level policy structure reduces the planning horizon for each policy but raises optimization challenges. For example, many HRL works require pretraining low-level skills (Singh, 1991; Kulkarni et al., 2016; Florensa et al., 2017; Riedmiller et al., 2018; Haarnoja et al., 2018a) or non-trivial algorithmic modifications to learn both modules jointly (Dayan & Hinton, 1993; Silver & Ciosek, 2012; Bacon et al., 2017; Vezhnevets et al., 2017; Nachum et al., 2018). On the contrary, our paradigm works with any policy architecture of enough representation power by imitating composite trajectories. Notably, such an idea of exploiting regularities on behavior instead of constraints on policy structures can be traced back to the "*Hierarchies of Machines*" model in 1998 (Parr & Russell, 1998).

Our approach is also related to the options framework (Sutton et al., 1999; Precup, 2000), which explicitly decomposes a complex task into a sequence of temporal abstractions, i.e., options. The options framework also assumes a hierarchical representation with an inter-option controller and a discrete set of reusable intra-option policies, which are typically non-trivial to obtain (Konidaris & Barto, 2009; Konidaris et al., 2012; Bagaria & Konidaris, 2020; Wulfmeier et al., 2020). SIR implicitly reuses previously learned "skills" via task reduction and distills all the composite strategies into a single policy. Besides, there are other works tackling different problems with conceptually similar

composition techniques. Probabilistic policy reuse (Fernández-Rebollo & Veloso, 2006; Fernández-Rebollo et al., 2010) executes a discrete set of previously learned policies for structural exploration. Fang et al. (2019) consider a dynamic goal setting without manipulable objects and compose an agent trajectory and a goal trajectory that collide as supervision. Dhiman et al. (2018) transfer previous knowledge to new goals in tabular cases by constraining value functions. Task reduction produces an expanding solved task space, which is conceptually related to life-long learning (Tessler et al., 2017) and open-ended learning (Wang et al., 2019).

Evidently, many recent advances have shown that with generic architectures, e.g., convolution (Le-Cun et al., 1995) and self-attention (Vaswani et al., 2017), simply training on massive data yields surprisingly intelligent agents for image generation (Chen et al., 2020), language modeling (Brown et al., 2020), dialogue (Adiwardana et al., 2020) and gaming (Schrittwieser et al., 2019; Baker et al., 2020). Hence, we suggest building intelligent RL agents via a similar principle, i.e., design a *diverse task spectrum* and then perform *supervised learning* on complex strategies.

The technical foundation of SIR is to use a universal value function and a goal conditioned policy (Kaelbling, 1993; Schaul et al., 2015), which has been successfully applied in various domains, including tackling sparse rewards (Andrychowicz et al., 2017), visual navigation (Zhu et al., 2017), planning (Srinivas et al., 2018), imitation learning (Ding et al., 2019) and continuous control (Nair et al., 2018). Besides, task reduction requires state-space search/planning over an approximate value function. Similar ideas of approximate state search have been also explored (Pathak et al., 2018; Eysenbach et al., 2019; Nasiriany et al., 2019; Ren et al., 2019; Nair & Finn, 2020). SIR accelerates learning by imitating its own behavior, which is typically called *self-imitation* (Oh et al., 2018; Ding et al., 2019; Lynch et al., 2019) or *self-supervision* (Nair et al., 2018; 2019; Pong et al., 2019; Ghosh et al., 2019). In this paper, we use techniques similar to Rajeswaran et al. (2018) and Oh et al. (2018), but in general, SIR does not rely on any particular imitation learning algorithm.

There are also related works on compositional problems of orthogonal interests, such as learning to smoothly compose skills (Lee et al., 2019), composing skills via embedding arithmetic (Sahni et al., 2017; Devin et al., 2019), composing skills in real-time (Peng et al., 2019), following instructions (Andreas et al., 2017; Jiang et al., 2019) and manually constructing "iterative-refinement" strategies to address supervised learning tasks (Nguyen et al., 2017; Hjelm et al., 2016; Chang et al., 2019).

Conceptually, our method is also related to *curriculum learning* (CL). Many automatic CL methods (Graves et al., 2017; Florensa et al., 2018; Matiisen et al., 2019; Portelas et al., 2019; Racaniere et al., 2020) also assume a task space and *explicitly* construct task curricula from easy cases to hard ones over the space. In our paradigm, task reduction reduces a hard task to a simpler solved task, so the reduction trace over the solved tasks can be viewed as an *implicit* task curriculum. Such a phenomenon of an implicitly ever-expanding solved subspace has been also reported in other works for tackling goal-conditioned RL problems (Warde-Farley et al., 2019; Lynch et al., 2019). Despite the conceptual similarity, there are major differences between SIR and CL. SIR assumes a given task sampler and primarily focuses on efficiently finding a policy to a given task via task decomposition. By contrast, CL focuses on task generation and typically treats policy learning as a black box. We also show in the experiment section that SIR is *complementary* to CL training and can solve a challenging sparse-reward stacking task (Sec. 5.2) that was unsolved by curriculum learning alone. We remark that it is also possible to extend SIR to a full CL method and leave it as future work.

## 3 PRELIMINARY AND NOTATIONS

We consider a multi-task reinforcement learning setting over a parameterized task space $\mathcal{T}$. Each task $T \in \mathcal{T}$ is represented as a goal-conditioned Markov Decision Process (MDP), $T = (\mathcal{S}, \mathcal{A}, s_T, g_T, r(s, a; g_T), P(s'|s, a; g_T))$, where $\mathcal{S}$ is the state space, $\mathcal{A}$ is the action space, $s_T, g_T \in \mathcal{S}$ are the initial state and the goal to reach, $r(s, a; g_T)$ is the reward function of task $T$, and $P(s'|s, a; g_T)$ is the transition probability. Particularly, we consider a *sparse reward* setting, i.e., $r(s, a; g_T) = \mathbf{I}\left[d(s, g_T) \leq \delta\right]$, where the agent only receives a non-zero reward when the goal is achieved w.r.t. some distance metric $d(\cdot, \cdot)$. We assume that the state space and the action space are shared across tasks, and different tasks share the same physical dynamics (i.e., $g_T$ only influences the termination condition of the transition probability $P$), so a task $T$ is determined by its initial state $s_T$ and goal $g_T$. We also assume a task sampler $C$, which samples training tasks from $\mathcal{T}$. $C$ can be simply a uniform sampler or something more intelligent. For evaluation, we are interested in the most

challenging tasks in $\mathcal{T}$. The learning agent is represented as a stochastic policy $\pi_\theta(a|s, g_T)$ conditioning on state $s$ and goal $g_T$ and parameterized by $\theta$. We aim to find the optimal parameter $\theta^\star$ such that policy $\pi_{\theta^\star}$ maximizes the expected accumulative reward across every task $T \in \mathcal{T}$ with a discounted factor $\gamma$, i.e., $\theta^\star = \arg\max_\theta \mathbb{E}\left[\sum_t \gamma^t r(s_t, a_t; g_T)\right]$ where $a_t \sim \pi_\theta(a|s_t, g_T)$. Correspondingly, we also learn a universal value function $V_\psi(s_t, g_T)$ for state $s_t$ and goal $g_T$ parameterized by $\psi$ w.r.t. $\pi_\theta$. The subscript $\theta$ in $\pi_\theta$ and $\psi$ in $V_\psi$ will be omitted for simplicity in the following content.

## 4 METHOD

### 4.1 TASK REDUCTION

The core idea of task reduction is to convert a hard-to-solve task $A \in \mathcal{T}$ to some task $B \in \mathcal{T}$ that the agent has learned to solve, so that the original task $A$ can be accomplished via composing two simpler tasks, i.e., (1) reducing task $A$ to $B$ and (2) solving the reduced task $B$. More formally, for a task $A$ with initial state $s_A$ and goal $g_A$ that the current policy $\pi$ cannot solve, we want to find a reduction target $B$ with initial state $s_B$ and goal $g_B$ such that (1) the task to reach state $s_B$ from $s_A$ is easy, and (2) $B$ has the same goal as $A$ (i.e., $g_B = g_A$) and solving task $B$ is easy. A natural measure of how *easy* a task is under $\pi$ is to evaluate the universal value function $V(\cdot)$. Hence, task reduction aims to find the best target state $s_B^\star$, which is the initial state of some task $B$, such that

$$s_B^\star = \arg\max_{s_B} V(s_A, s_B) \oplus V(s_B, g_A) := \arg\max_{s_B} V(s_A, s_B, g_A), \tag{1}$$

where $\oplus$ is some composition operator and $V(s_A, s_B, g_A)$ denotes the composite value w.r.t a particular target state $s_B$. There are multiple choices for $\oplus$, including minimum value, average or product (due to 0/1 sparse rewards). Empirically, taking the product of two values leads to the best practical performance. Thus, we assume $V(s_A, s_B, g_A) := V(s_A, s_B) \cdot V(s_B, g_A)$ in this paper.

With the optimal target state $s_B^\star$ obtained, the agent can perform task reduction by executing the policy $\pi(a|s_A, s_B^\star)$ from the initial state $s_A$. If this policy execution leads to some successful state $s_B'$ (i.e., within a threshold to $s_B^\star$), the agent can continue executing $\pi(a|s_B', g_A)$ from state $s_B'$ towards the ultimate goal $g_A$. Note that it may not always be possible to apply task reduction successfully since either of these two policy executions can be a failure. There are two causes of failure: (i) the approximation error of the value function; (ii) task $A$ is merely too hard for the current policy. For cause (i), the error is intrinsic and cannot be even realized until we actually execute the policy in the environment; while for cause (ii), we are able to improve sample efficiency by only performing task reduction when it is *likely* to succeed, e.g., when $V(s_A, s_B^\star, g_A)$ exceeds some threshold $\sigma$.

### 4.2 SELF-IMITATION

Once task reduction is successfully applied to a hard task $A$ via a target state $s_B^\star$, we obtain two trajectories, one for reducing $A$ to $B$ with final state $s_B'$, and the other for the reduced task $B$ starting at $s_B'$. Concatenating these two trajectories yields a successful demonstration for the original task $A$. Therefore, by continuously collecting successful trajectories produced by task reduction, we can perform imitation learning on these self-demonstrations. Suppose demonstration dataset is denoted by $\mathcal{D}^{il} = \{(s, a, g)\}$. We utilize a weighted imitation learning objective (Oh et al., 2018) as follows:

$$L^{il}(\theta; \mathcal{D}^{il}) = \mathbb{E}_{(s,a,g) \in \mathcal{D}^{il}} \left[\log \pi(a|s, g) A(s, g)_+\right], \tag{2}$$

where $A(s, g)_+$ denotes the clipped advantage, i.e., $\max(0, R - V(s, g))$. Self-imitation allows the agent to more efficiently solve harder tasks by explicitly using past experiences on easy tasks.

### 4.3 SIR: SELF-IMITATION VIA REDUCTION

We present SIR by combining deep RL and self-imitation on reduction demonstrations. SIR is compatible with both off-policy and on-policy RL. We use soft actor-critic (Haarnoja et al., 2018b) with hindsight experience replay (Andrychowicz et al., 2017) (SAC) and proximal policy gradient (PPO) (Schulman et al., 2017) respectively in this paper. In general, an RL algorithm trains an actor and a critic with rollout data $\mathcal{D}^{rl}$ from the environment or the replay buffer by optimizing algorithm-specific actor loss $L^{actor}(\theta; \mathcal{D}^{rl})$ and critic loss $L^{critic}(\psi; \mathcal{D}^{rl})$.

For policy learning, since our reduction demonstrations $\mathcal{D}^{il}$ are largely off-policy, it is straightforward to incorporate $\mathcal{D}^{il}$ in the off-policy setting: we can simply aggregate the demonstrations into the experience replay and jointly optimize the self-imitation loss and the policy learning loss by

$$L^{\text{SIR}}(\theta; \mathcal{D}^{rl}, \mathcal{D}^{il}) = L^{actor}(\theta; \mathcal{D}^{rl} \cup \mathcal{D}^{il}) + \beta L^{il}(\theta; \mathcal{D}^{rl} \cup \mathcal{D}^{il}).$$

While in the on-policy setting, we only perform imitation on demonstrations. Since the size of $\mathcal{D}^{il}$ is typically much smaller than $\mathcal{D}^{rl}$, we utilize the following per-instance loss to prevent overfit on $\mathcal{D}^{il}$:

$$L^{\text{SIR}}(\theta; \mathcal{D}^{rl}, \mathcal{D}^{il}) = \frac{\|\mathcal{D}^{rl}\| L^{actor}(\theta; \mathcal{D}^{rl}) + \|\mathcal{D}^{il}\| L^{il}(\theta; \mathcal{D}^{il})}{\|\mathcal{D}^{rl} \cup \mathcal{D}^{il}\|}.$$

For critic learning, since the objective is consistent with both data sources, we can simply optimize $L^{critic}(\psi; \mathcal{D}^{rl} \cup \mathcal{D}^{il})$. The proposed algorithm is summarized in Algo. 1.

### 4.4 Implementation and Extensions

**Task Space:** We primarily consider *object-based* state space, where a goal state is determined by whether an object reaches some desired location. We also extend our method to image input by learning a low-dimensional latent representation (Higgins et al., 2017b) and then perform task reduction over the latent states (see Sec. 5.3). We do not consider strong partial-observability (Pinto et al., 2018) in this paper and leave it as future work.

**Reduction-Target Search:** In our implementation, we simply perform a random search to find the top-2 candidates for $s_B^\star$ in Eq. (1) and try both of them. We also use the cross-entropy method (De Boer et al., 2005) for efficient planning over the latent space with image input (Sec. 5.3). Alternative planning methods can be also applied, such as Bayesian optimization (Snoek et al., 2012; Swersky et al., 2014), gradient descent (Srinivas et al., 2018), or learning a neural planner (Wang et al., 2018; Nair et al., 2019). We emphasize that the reduction search is performed completely on the learned value function, which does not require *any environment interactions*.

---

**Algorithm 1:** Self-Imitation via Reduction

**initialize** $\pi_\theta$, $V_\psi$; $\mathcal{D}^{rl} \leftarrow \emptyset$, $\mathcal{D}^{il} \leftarrow \emptyset$;
**repeat**
    $\tau \leftarrow$ trajectory by $\pi_\theta$ in task $A$ from $C$;
    $\mathcal{D}^{rl} \leftarrow \mathcal{D}^{rl} + \{\tau\}$;
    **if** $\tau$ *is a failure* **then**
        find target state $s_B^\star$ w.r.t. Eq. (1);
        **if** $V(s_A, s_B^\star, g_A) > \sigma$ **then**
            $\tau' \leftarrow$ trajectory by reduction;
            **if** $\tau'$ *is a success* **then**
                $\mathcal{D}^{il} \leftarrow \mathcal{D}^{il} + \{\tau'\}$;
    **if** *enough new data collected* **then**
        $\theta \leftarrow \theta + \alpha \nabla L^{\text{SIR}}(\theta; \mathcal{D}^{rl}, \mathcal{D}^{il})$;
        $\psi \leftarrow \psi + \alpha \nabla L^{critic}(\psi; \mathcal{D}^{rl} \cup \mathcal{D}^{il})$;
**until** *enough iterations*;
**return** $\pi_\theta$, $V_\psi$;

---

Particularly in the object-centric state space, an extremely effective strategy for reducing the search space is to constrain the target $s_B$ to differ from the current state $s_A$ by *just a few objects* (e.g., 1 object in our applications). Due to the compositionality over objects, the agent can still learn arbitrarily complex behavior recursively throughout the training. When performing task reduction in training, SIR explicitly resets the initial state and goal of the current task, which is a very common setting in curriculum learning and goal-conditioned policy learning. It is possible to avoid resets by archiving solutions to solved tasks to approximate a reduction trajectory. We leave it as future work.

**Estimating $\sigma$:** We use a static $\sigma$ for SAC. For PPO, we empirically observe that, as training proceeds, the output scale of the learned value function keeps increasing. Hence, we maintain an online estimate for $\sigma$, i.e., a running average of the composite values over all the selected reduction targets.

**Imitation Learning:** Note that there can be multiple feasible reduction targets for a particular task. Hence, the self-demonstrations can be highly multi-modal. This issue is out of the scope of this paper but it can be possibly addressed through some recent advances in imitation learning (Ho & Ermon, 2016; Guo et al., 2019; Lynch et al., 2019).

## 5 Experiments

We compare SIR with baselines without task reduction. Experiments are presented in 3 different environments simulated in MuJoCo (Todorov et al., 2012) engine: a robotic-hand pushing scenario

with a wall opening (denoted by "*Push*"), a robotic-gripper stacking scenario (denoted by "*Stack*"), and a 2D particle-based maze scenario with a much larger environment space (denoted by "*Maze*"). All of the scenarios allow random initial and goal states so as to form a continuously parameterized task space. All of the problems are fully-observable with ***0/1 sparse rewards*** and are able to restart from a given state when performing task reduction. All the training results are repeated over 3 random seeds. We empirically notice that SAC has superior performance in robotics tasks but takes significantly longer *wall-clock time* than PPO in the maze navigation tasks (see Fig. 17 in Appx. C). Hence, we use SAC in *Push* and *Stack* while PPO in *Maze* for RL training in all the algorithms if not otherwise stated. We remark that we also additionally report SAC results for a simplified *Maze* task in Appx. C (see Fig. 18). We primarily consider state-based observations and additionally explore the extension to image input in a *Maze* scenario. More details and results can be found in the appendices.

## 5.1 *Push* SCENARIO

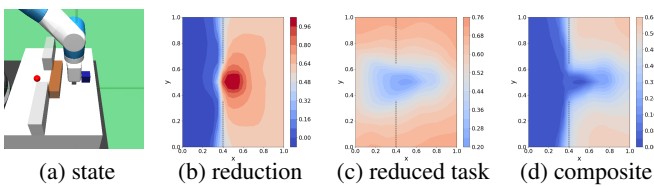

(a) state      (b) reduction      (c) reduced task      (d) composite

Figure 2: Heatmap of learned value function. (a) current state; (b) value for moving the elongated box elsewhere; (c) value for moving the cubic box to goal w.r.t. different elong. box pos.; (d) composite value for different reduction targets: moving elong. box away from door yields high values for task reduction.

|           | Hard  | Rand. |
|-----------|-------|-------|
| SAC only  | 0.406 | 0.771 |
| SAC + TR  | 0.764 | 0.772 |

Table 1: Task Reduction (TR) as a test-time planner. We evaluate the success rates on test tasks when additional TR steps are allowed in the execution of the SAC-trained policy.

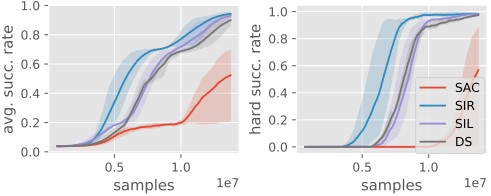

Figure 3: Avg. success rate w.r.t. samples over current task distribution (left) and in hard cases (right) in the *Push* scenario.

As shown in Fig. 1 (Sec. 1), there are two objects on the table, a cubic box (blue) and an elongated box (brown). The robot needs to push a specified object to a goal location (red). The table is separated into two connected regions by a fixed wall with an open door in the middle. The initial object positions, goal location, robot hand location, and which object to push are randomly reset per episode.

**Learned Value Function:** We retrieve a partially trained value function during learning and illustrate how task reduction is performed over it in Fig. 2. Consider a hard task shown in Fig. 2a where the door is blocked by the elongated box and the agent needs to move the cubic box to the goal on the other side. A possible search space is the easier tasks with the position of the elongated box perturbed. We demonstrate the learned values for moving the elongated box to different goal positions in Fig. 2b, where we observe consistently lower values for faraway goals. Similarly, Fig. 2c visualizes the values for all possible reduced tasks, i.e., moving the cubic box to the goal with the elongated box placed in different places. We can observe higher values when the elongated box is away from the door. Fig. 2d shows the composite values w.r.t. different target states, where the reduction search is actually performed. We can clearly see two modes on both sides of the door.

**Task Reduction as a Planner:** Task reduction can serve as a test-time planner by searching for a reduction target that possibly yields a higher composite value than direct policy execution. We use a half-trained policy and value function by SAC and test in a fixed set of 1000 random tasks. Since random sampling rarely produces hard cases, we also evaluate in 1000 *hard tasks* where the elongated box blocks the door and separates the cubic box and goal. The results are in Table. 1, where task reduction effectively boosts performances in hard cases, which require effective planning.

**Performances:** SIR is compared with three different baselines: SAC; SAC with self-imitation learning (SIL), which does not run task reduction and only imitates past successful trajectories directly generated by policy execution; SAC with a dense reward (DS), i.e., a potential-based distance reward from object to the goal. We evaluate the average success rate over the entire task distribution

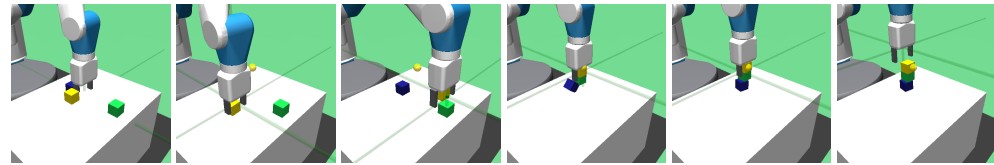

Figure 4: SIR-learned strategy for stacking 3 boxes in the *Stack* scenario. The yellow spot in the air is the goal, which requires a yellow box to be stacked to reach it.

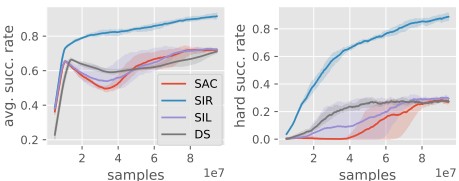

| # Box | 1 | 2 | 3 |
|-------|---|---|---|
| SAC | **0.96±0.01** | 0.94±0.03 | 0.25±0.03 |
| SIL | 0.89±0.06 | 0.96±0.03 | 0.28±0.02 |
| DS | 0.93±0.02 | 0.67±0.15 | 0.27±0.01 |
| SIR | **0.96±0.03** | **0.98±0.01** | **0.89±0.03** |

Figure 5: Training results in the *Stack* scenario with at most 3 boxes. We evaluate success rate across training curriculum (left) and in hard cases, i.e., stacking 3 boxes (right).

Table 2: Average success rate and standard deviation of different algorithms for stacking different number of boxes evaluated at the same training samples.

as well as the hard-case success rate (i.e., with a blocked door) in Fig. 3, where SIR achieves the best overall performances, particularly in hard cases. We emphasize that for SIR, we include all the discarded samples due to failed task reductions in the plots. Since the only difference between SIR and SIL is the use of task reduction, our improvement over SIL suggests that task reduction is effective for those compositional challenges while burdened with minimal overall overhead.

## 5.2 *Stack* SCENARIO

Next, we consider a substantially more challenging task in a similar environment to *Push*: there is a robotics gripper and at most 3 cubic boxes with different colors on the table. The goal is to stack a tower using these boxes such that a specified cubic box remains stable at a desired goal position even with gripper fingers left apart. Therefore, if the goal is in the air, the agent must learn to first stack non-target boxes in the bottom, to which there are even multiple valid solutions, before it can finally put the target box on the top. See Fig. 4 for an example of stacking the yellow box towards the goal position (i.e, the yellow spot in the air). Each episode starts with a random number of boxes with random initial locations and a colored goal that is viable to reach via stacking.

This task is extremely challenging, so we (1) introduce an auxiliary training task, *pick-and-place*, whose goal is simply relocating a specified box to the goal position, and (2) carefully design a training curriculum for all methods by gradually decreasing the ratio of pick-and-place tasks from 70% to 30% throughout the training. For evaluation, a task is considered as a hard case if all the 3 boxes are required to be stacked together.

We show the training curves as well as the hard-case success rate for SIR and other methods in Fig. 5. SIR significantly outperforms all the baselines, especially in the hard cases, i.e., stacking all the 3 boxes. Table. 2 illustrates the detailed success rate of different methods for stacking 1, 2 and 3 boxes. SIR reliably solves all the cases while other methods without task reduction struggle to discover the desired compositional strategy to solve the hardest cases. Fig. 4 visualizes a successful trajectory by SIR for stacking 3 boxes from scratch, where the goal is to stack the yellow box to reach the goal spot in the air. Interestingly, the agent even reliably discovers a non-trivial strategy: it first puts the yellow box on the top of the green box to form a tower of two stories, and then picks these two boxes jointly, and finally places this tower on top of the blue box to finish the task.

## 5.3 *Maze* SCENARIO

In this scenario, we focus on a navigation problem with a particle-based agent in a large 2D maze, named "*4-Room maze*". The maze is separated into 4 connected rooms by three fixed walls and each wall has an open door in the middle. There are 4 objects in the maze, including 3 elongated boxes

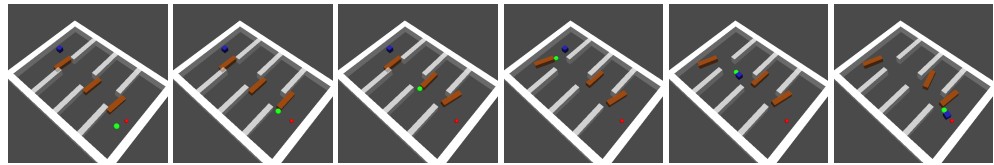

Figure 6: SIR-learned strategy for a hard case in the *4-Room maze* scenario. The agent(green) is tasked to move the blue box to the target(red). All three doors are blocked by elongated boxes, so the agent needs to clear them before pushing the blue box towards the target.

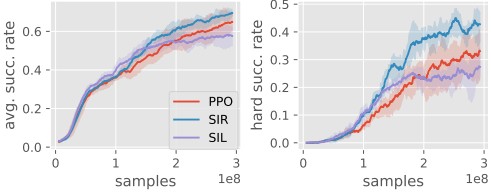

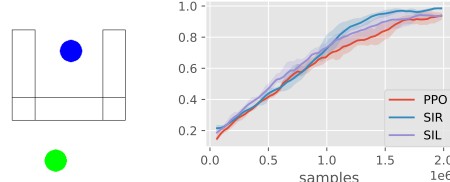

Figure 7: Results in *4-Room maze*. Left: success rate in the training task distribution; Right: success rate evaluated only in the hard cases.

Figure 8: Navigation in *U-Wall maze* with image input. Left: task visualization; Right: avg. training success rates w.r.t. samples.

(brown) and 1 cubic box (blue). The task for the agent (green) is to move a specified object to a goal position (red). We consider a case hard when (i) the agent needs to move the cubic box to a different room, and (ii) at least one door is blocked by the elongated boxes.

Figure 6 visualizes a particularly challenging situation: the agent needs to move the cubic box to the goal but all three doors are precisely blocked by the elongated boxes. To solve this task, the agent must perturb all the elongated boxes to clear the doors before moving the cubic box. As is shown in Fig. 6, the agent trained by SIR successfully learns this non-trivial strategy and finishes the task.

We adopt PPO for training here and compare SIR to baselines in terms of average success rate over both the training task distribution and the hard tasks only. The results are shown in Fig. 7. SIR outperforms both PPO and SIL baselines with a clear margin, especially when evaluated on hard cases. Note that although SIL learns slightly more efficiently at the beginning of training, it gets stuck later and does not make sufficient progress towards the most challenging cases requiring complex compositional strategies to solve.

**Extension to visual domain:** We consider extending SIR to the setting of image input by first pre-training a $\beta$-VAE (Higgins et al., 2017a) to project the high-dimensional image to a low-dimensional latent vector and then perform task reduction over the latent space. We train the $\beta$-VAE using random samples from the environment and apply the cross-entropy method (CEM) (De Boer et al., 2005) to search for the reduction targets. We experiment in a sparse-reward variant of the *U-Wall maze* task adapted from Nasiriany et al. (2019), where the agent (blue) needs to navigate to the goal (green) by turning around the wall (boundary plotted in black). Task visualization and training results are shown in Fig. 8, where SIR still achieves the best performance compared with PPO and SIL.

**Comparison with HRL:** We also compare SIR with two baselines with hierarchical policy structures, i.e., an off-policy HRL algorithm HIRO (Nachum et al., 2018) and an option discovery algorithm DSC (Bagaria & Konidaris, 2020). Since these two HRL baselines are both based on off-policy learning, we use SAC as our training algorithm in SIR for a fair comparison. In addition, since both

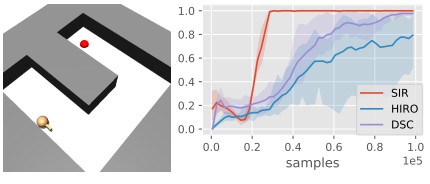

(a) *U-Shape maze*, fixed goal

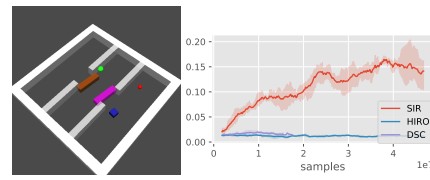

(b) Box pushing in *3-Room maze*, fixed goal

Figure 9: Comparison of HRL baselines and SIR in the fixed-goal *Maze* scenarios.

two baselines also assume a fixed goal position, we experiment in two fixed-goal *Maze* scenarios. We first conduct a sanity check in a simple *U-Shape maze* without any boxes (Fig. 9a), where the agent (yellow) is expected to propel itself towards the goal (red). Then we experiment in a *3-Room maze* (Fig. 9b), which is a simplified version of the previously described *4-Room maze*. The results are shown in Fig. 9. Although HIRO and DSC learn to solve the easy *U-shape maze*, they hardly make any progress in the more difficult *3-Room maze*, which requires a non-trivial compositional strategy. Note that DSC is extremely slow in *3-Room maze*, so we terminate it after *5 days* of training. In contrast, even without any policy hierarchies, SIR learns significantly faster in both two scenarios.

**Comparison with automatic curriculum learning:** As discussed at the end of Sec. 2, SIR *implicitly* constructs a task curriculum by first solving easy tasks and then reducing a hard task to a solved one. Hence, we also conduct experiments with curriculum learning (CL) methods. We compare SIR using a *uniform task sampler* with standard PPO learning with Goal-GAN (Florensa et al., 2018), an automatic CL approach, as well as a manually designed training curriculum, which gradually increases the distance between the goal and the start position. Since GoalGAN assumes a fixed initial position for the agent while only specifies the goal position, we consider the unmodified navigation problem used in Florensa et al.

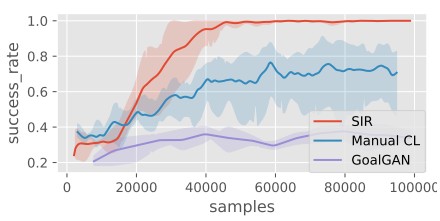

Figure 10: Comparison with curriculum learning methods in the *U-Shape maze (fixed goal)*. We consider SIR with a uniform task sampler and standard PPO with GoalGAN and a manually designed curriculum.

(2018) for a fair comparison. The results are shown in Fig. 10. Even with a uniform task sampler, SIR significantly outperforms the CL baselines that do not leverage the task reduction technique. We want to remark that GoalGAN eventually solves the task with $10^6$ samples since it takes a large number of samples to train an effective GAN for task sampling. By contrast, SIR directly works with a uniform task sampler and implicitly learns from easy to hard in a much lightweight fashion. Finally, we emphasize that SIR is also complementary to curriculum learning methods. As we previously show in Sec. 5.2, combining SIR and a manually-designed training curriculum solves a challenging stacking problem, which is unsolved by solely running curriculum learning.

# 6 CONCLUSION

This paper presents a novel learning paradigm SIR for solving complex RL problems. SIR utilizes task reduction to leverage compositional structure in the task space and performs imitation learning to effectively tackle successively more challenging tasks. We show that an RL agent trained by SIR can efficiently learn sophisticated strategies in challenging sparse-reward continuous-control problems. Task reduction suggests a simple but effective way, i.e., supervised learning on composite trajectories with a generic policy representation, to incorporate inductive bias into policy learning, which, we hope, could bring valuable insights to the community towards a wider range of challenges.

### ACKNOWLEDGMENTS

Co-author Wang was supported, in part, by gifts from Qualcomm and TuSimple.

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

# A ENVIRONMENT DETAILS

## A.1 PUSH SCENARIO

The robot is placed beside a 50cm × 70cm table with an elongated box and a cubic box on it. The initial position of the two boxes, the projection of the initial gripper position onto the table plane, and the goal position are sampled from a 30cm × 30cm square on the table. The gripper is initialized at a fixed height above the table.

**Observation space** The observation vector is a concatenation of *state*, *achieved goal* and *desired goal*. *State* consists of the position and velocity of the gripper, the cubic box and the elongated box, as well as the state and velocity of the gripper fingers. *Desired goal* vector consists of a position coordinate and a one-hot vector indicating the identity of the target object. *Achieved goal* contains the current position of the target object and its corresponding one-hot index vector.

**Action space** The action is 4-dimensional, with the first 3 elements indicating the desired position shift of the end-effector and the last element controlling the gripper fingers (locked in this scenario).

**Hard case configuration** Our training distribution is a mixture of easy and hard cases. In easy cases, positions of the gripper, the boxes and the goal are all uniformly sampled. The index of the target object is also uniformly sampled from total number of objects. In hard cases, the position of the gripper, the position of the cubic box, and the target object index are still uniformly sampled, but the elongated box is initialized at a fixed location that blocks the door open of the wall. The goal is on the opposite side of the wall if the target object is the cubic box.

## A.2 STACK SCENARIO

The physical setup of this scenario is much the same as that in *Push* scenario, except that there is no wall on the table. At the beginning of each episode, the number of boxes on the table, the goal position and the target object index are sampled from a designed task curriculum. As explained in the main paper, there are two types of tasks, stacking and pick-and-place in the curriculum, and the type of task is also reset in each episode.

The $(x, y)$ coordinate of the goal is uniformly sampled from a 30cm × 30cm square on the table, while goal height $g_z$ is initialized depending on the type of task of the current episode: for pick-and-place tasks, $g_z \sim \text{table\_height} + U(0, 0.45)$ and for stacking tasks, $g_z = \text{table\_height} + N \cdot \text{box\_height}$, where $N \in [1, N_{max}]$ denotes the number of boxes.

**Observation space** The observation vector shares the same structure with *Push* scenario, but includes another one-hot vector indicating the type of the task.

**Action space** The action vector is the same as that in *Push* scenario.

**Hard case configuration** The hard cases we are most interested in are stacking $N_{max}$ boxes so that the target box reaches the goal position with the gripper fingers left apart.

## A.3 MAZE SCENARIO

This scenario deals with a 2D maze. The agent (green) can only apply force to itself in the directions parallel to the floor plane. The position of the agent and the goal are both uniformly sampled from the collision-free space in the maze.

**Observation space** The structure of observation space is generally the same as the *Push* scenario except that the state and velocity of gripper fingers are not included in the *state* vector since there is no gripper in *Maze* scenarios. In *U-Shape Maze*, the one-hot target object indicator is omitted since there is no object except the agent itself. For experiments in the visual domain, RGB images of shape $48 \times 48$ are directly used as observations.

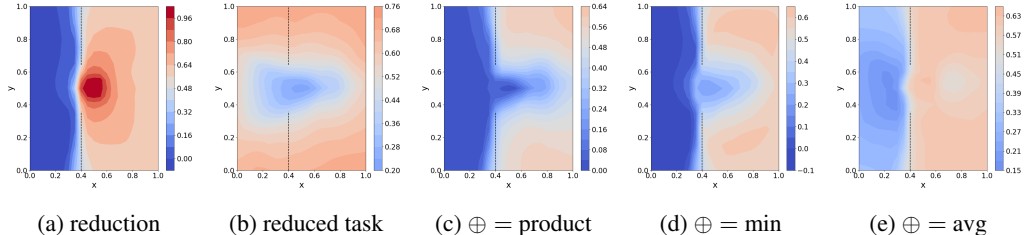

Figure 11: Heatmap of value function with different composition operators. (a) value for moving elongated box elsewhere, (b) value for moving cubic box to goal w.r.t. different elongated box position, (c) composite value with product operator, (d) composite value with min operator, (e) composite value with average operator.

**Action space**   The action is a 2-dimensional vector representing the force applied to the agent.

**Hard case configuration**   There is no manually designed hard case for *U-Shape Maze* and *U-Wall Maze*. In hard cases of *4-Room maze*, a random number of elongated boxes block the doors on. The goal is forced to initialize in a different room from that the cubic box is in. For hard cases of *3-Room Maze*, each elongated box blocks a door. More precisely, the rooms that elongated boxes are in are determined by the agent's position: if the agent is in the leftmost room, the elongated boxes are placed to the left of each door; if the agent is in the middle room, the elongated boxes are also placed in the middle room and each blocks a door; if the agent is in the rightmost room, the elongated boxes are placed to the right of each door.

**Fixed goal location**   In the fixed goal version of *3-Room maze*, the goal is located at the same spot in the rightmost room for each episode. In *U-Shape Maze*, the goal is always at the end of the maze.

## B   TRAINING DETAILS

**Composition operator**   As is mentioned in Sec. 4.1, there are multiple choices for the composition operator given two values (Eq. (1)). In addition to the one we used in SIR, i.e., product operator, we also visualize the composite values derived by other operators, including minimum (i.e., "min", taking the smaller value of the two) and average (i.e., take the mean value) in Fig. 11.

Besides, note that due to value approximation error, even if the reward is 0/1, the output of a learned value function may not perfectly lie between 0 and 1. So, in order to apply the "product" operator well in practice, we normalize the value functions before they are actually composed. Specifically, the composite values are computed as follow:

$$V(s_A, s_{B_i}; g_A, \oplus = \text{product}) = \frac{V(s_A, s_{B_i}) - \min_j V(s_A, s_{B_j})}{\max_j V(s_A, s_{B_j}) - \min_j V(s_A, s_{B_j})}$$
$$\frac{V(s_{B_i}, g_A) - \min_j V(s_{B_j}, g_A)}{\max_j V(s_{B_j}, g_A) - \min_j V(s_{B_j}, g_A)}$$

$$V(s_A, s_{B_i}; g_A, \oplus = \text{min}) = \min(V(s_A, s_{B_i}), V(s_{B_i}, g_A))$$

$$V(s_A, s_{B_i}; g_A, \oplus = \text{avg}) = \frac{V(s_A, s_{B_i}) + V(s_{B_i}, g_A)}{2}$$

Empirically, the "product" operator is the best among all candidates, since it most clearly demonstrates two high-value modes on both sides of the door and low composite values at the door.

**Estimating $\sigma$**   When combining SIR with SAC baseline, the output of value function converges to a stable range after the initial warm-up phase (see Fig. 12a). Therefore it suffices to use a static high threshold (set to 0.9 in *Push* scenario with mixed task sampler and 1 in all other cases) to

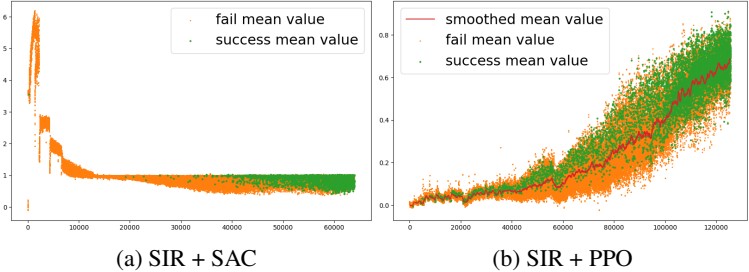

(a) SIR + SAC                      (b) SIR + PPO

Figure 12: Values of all the reduction targets with different baseline algorithms. Green: successful targets; Orange: failed targets; Red (in the right plot): running average of values.

discard reduction targets in the initial value overestimation phase and a static low threshold $\sigma$ to filter out reduction targets with low values that are not likely to succeed. When training SIR with PPO baseline, We empirically observe that the scale of value function keeps increasing as training proceeds, therefore a static filtering threshold is not appropriate here. Hence we maintain an online estimation of the filtering threshold by keeping a running average of composite values of all the selected reduction targets. As is shown in Fig. 12b, the successful reduction targets typically have higher values than average.

**Designed task curriculum in Stack scenario** The initial positions of all the boxes are generated as follows: select $M$ $(0 \leq M \leq N - 1)$ boxes from the total $N$ boxes, and stack them under the goal, then initialize other boxes randomly over a square on the table. The ratio of stacking and pick-and-place tasks is adapted according to the agent's learning progress. We keep 70% chance of sampling pick-and-place tasks until the agent achieves a success rate of 0.8 when evaluating on pick-and-place tasks, then gradually decrease the ratio to 30%. Note that such task curriculum is applied for all the experimented algorithms in *Stack* scenario.

**Baseline implementation** We utilize prioritized experience replay when working with off-policy RL algorithms. Two separate replay buffers $\mathcal{D}^{rl}, \mathcal{D}^{il}$ are prepared for rollouts generated by direct policy execution and composite task reduction. We compute priority as $p = |r + \gamma V - Q|$ for each instance when storing transitions into the buffers. When sampling from $\mathcal{D}^{rl} \cup \mathcal{D}^{il}$, each element can be sampled with probability proportional to $p^{\alpha}$. The replay buffers maintain a fixed finite number of samples and data are flushed following the first-in-first-out manner.

As for on-policy baselines, data in $\mathcal{D}^{rl}$ are discarded after every iteration. $\mathcal{D}^{il}$ keeps track of a recent history of task reduction demonstration. As the training proceeds, we get better policies and new demonstrations with higher quality. Some stale demonstrations may not be beneficial anymore (e.g., early accidental successes by random exploration). However, the advantage clipping scheme in Eq. (2) may not effectively eliminate undesired stale data: in the sparse reward setting, since all the demonstrations are successful trajectories, the advantage will be hardly negative[1]. Thus, we maintain the dataset $\mathcal{D}^{il}$ as a buffer which keeps every trajectory for a fixed number of iterations after it was generated.

**Network architecture** In *Push* scenario, we simply use MLPs with 256 hidden units per layer to map the input observations to actions or values.

In *Stack* and *Maze* scenario, since there are more objects in observations, we adopt an attention-based feature extractor to get representation invariant to the number of objects. The original observation vector can be rewritten as $[o_{self}, o_1, o_2, \cdots, o_N]$, where $o_{self}$ contains the state of the agent (robot or particle) and the position of the desired goal, $o_i (1 \leq i \leq N)$ represents the state of each object. We then apply MLP to get observation embeddings $f(o_{self}), g(o_i, 1 \leq i \leq N)$. The attention embedding $e$ of objects is then computed by attending $f(o_{self})$ over $g(o_i)$:

$$e = \sum_i w_i g(o_i), w_i = \text{softmax}(f(o_{self}) \cdot g(o_i)), 1 \leq i \leq N. \tag{3}$$

---

[1]Though, it can be negative in theory due to discounted factor.

|                   | Push | Stack        |
|-------------------|------|--------------|
| #workers          | 32   | 32           |
| replay buffer size| 1e5  | 1e5          |
| batch size        | 256  | 256          |
| $\gamma$          | 0.98 | 0.98         |
| learning rate     | 3e-4 | 3e-4         |
| $\sigma$          | 0.7  | 0.5          |
| total timesteps   | 1.5e7| 1e8 (3 boxes)|
|                   |      | 2.5e7 (2 boxes)|

Table 3: Hyperparameters for SAC-based experiments in different scenarios.

|                | Push | Maze |
|----------------|------|------|
| #minibatches   | 32   | 32   |
| $\gamma$       | 0.99 | 0.99 |
| #opt epochs    | 10   | 10   |
| learning rate  | 3e-4 | 3e-4 |
| #workers       | 32   | 64   |
| #steps per iter| 2048 | 8192 |
| demo reuse     | 8    | 4    |
| total timesteps| 5e7  | 3e8  |

Table 4: Hyperparameters for PPO-based experiments in different scenarios.

|                          | Pure RL | SIL    | DS    | SIR    |
|--------------------------|---------|--------|-------|--------|
| Push Unif.               | 0.853   | **0.952** | 0.891 | 0.938  |
| Push Mixed 30%           | 0.523   | 0.930  | 0.900 | **0.942** |
| Stack (at most 2 boxes)  | 0.965   | 0.966  | 0.908 | **0.980** |
| Stack (at most 3 boxes)  | 0.721   | 0.722  | 0.711 | **0.895** |
| 4-Room maze              | 0.649   | 0.576  | 0.645 | **0.695** |

Table 5: Final average success rate over the training task distribution.

Finally $e$ is concatenated with $f(o_{self})$ and fed into another MLP to produce actions or values.

**Hyperparameters**   We summarize hyperparameters of off-policy and on-policy algorithms in each scenario in Table 3 and Table 4.

## C   MORE RESULTS

**Success rate statistics**   The success rates over the training task distribution and on hard cases evaluated at the end of training for all the experimented algorithms are listed in Table 5 and Table 6. All the numbers are averaged over 3 random seeds. The proposed SIR generally achieves the best result across the board.

Particularly in *Stack* scenario with 3 boxes, we provide more detailed hard-case success rates from all kinds of initial configurations for stacking 3 boxes (explained in Sec. B ) in Table. 7. Although all the algorithms succeed from the initial states with 2 boxes already stacked by simply stacking the target box on top of the existing tower, only SIR can solve more compositional situations with 1 or no box stacked in the beginning, which require strategic manipulation of other non-target boxes before finishing the tasks.

**Additional results in Push scenario**   In addition to the results presented in the main paper, which is trained on a mixture of 30% hard cases and 70% uniformly sampled cases, we also show the training curves with a uniform task sampler in Fig. 13. Furthermore, we repeat these experiments

|                          | Pure RL | SIL   | DS      | SIR     |
|--------------------------|---------|-------|---------|---------|
| Push Unif.               | 0.449   | 0.816 | 0.598   | **0.939** |
| Push Mixed 30%           | 0.568   | 0.979 | 0.975   | **0.983** |
| Stack (at most 2 boxes)  | 0.927   | 0.954 | 0.759   | **0.988** |
| Stack (at most 3 boxes)  | 0.270   | 0.294 | 0.275   | **0.846** |
| 4-Room maze              | 0.331   | 0.274 | **0.512** | 0.427   |

Table 6: Final hard-case success rate.

|              | Pure RL | SIL   | DS    | SIR       |
|--------------|---------|-------|-------|-----------|
| 2 boxes stacked | 0.851 | 0.899 | 0.889 | **0.972** |
| 1 box stacked   | 0.000 | 0.000 | 0.000 | **0.911** |
| From scratch    | 0.000 | 0.000 | 0.000 | **0.716** |

Table 7: Hard-case success rate in *Stack* scenario evaluated in all kinds of initial configurations for stacking 3 boxes in the task curriculum.

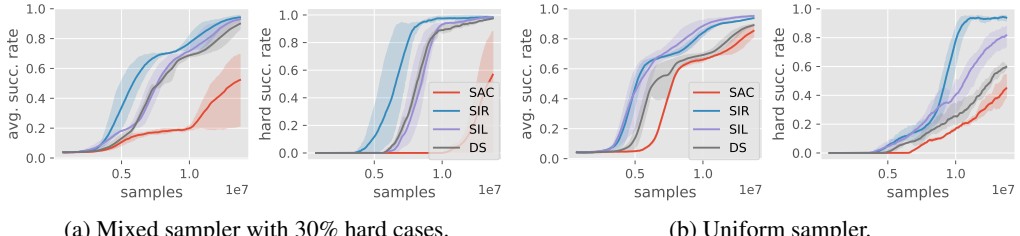

(a) Mixed sampler with 30% hard cases.  (b) Uniform sampler.

Figure 13: Training results of SAC-based algorithms in the *Push* scenario with 2 different task samplers. In each case, we show avg. success rate w.r.t. samples over current task distribution (left) and in hard cases (right).

with PPO-based algorithms as shown in Fig. 14. SIR achieves generally the best result compared with other baselines in all the settings, especially in hard cases which require compositional policies. It also illustrates that SIR is agnostic to baseline RL algorithms.

**Additional results in Stack scenario**   In *Stack* scenario, we also provide results trained with a maximum of two boxes. SIR is compared with other three baselines in Fig. 16. Our method is significantly more sample efficient especially when learning to stack.

**Additional results in Maze scenario**   We empirically find that SAC runs a magnitude slower in wall-clock time than PPO in *n-Room maze* as shown in Figure 17, therefore we use PPO-based algorithms in *4-Room maze* experiments in the main paper. The result with SAC-based algorithms in *3-Room maze* scenario is shown in Figure 18. SIR still outperforms SAC and SIL baselines in terms of average success rate over both the training task distribution and the hard tasks only.

We also include the DS reward explained in *Push* scenario as a baseline for comparison in Figure 19. Interestingly, DS surpasses all other methods when evaluated in hard cases only in the later training phase since it discovers how to navigate around the walls and push the blocking box away at the same time and remembers this policy.

**Episode length**   We are also interested in if SIR-learned policy can efficiently solve tasks besides the success rate reported in the main paper. Hence we test trained policies with different algorithms

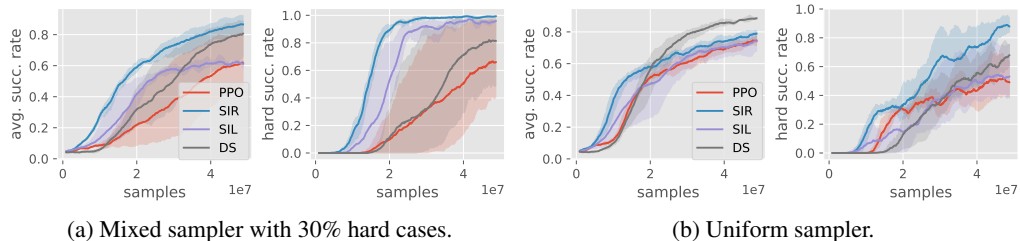

(a) Mixed sampler with 30% hard cases.  (b) Uniform sampler.

Figure 14: Training results of PPO-based algorithms in the *Push* scenario with 2 different task samplers. In each case, we show avg. success rate w.r.t. samples over current task distribution (left) and in hard cases (right).

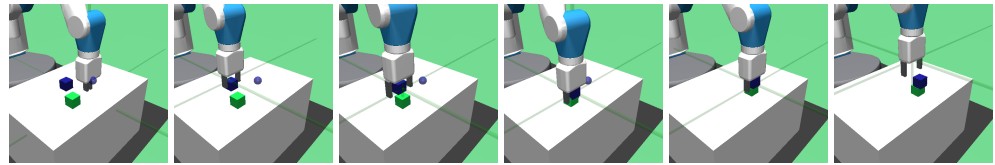

Figure 15: SIR-learned strategy for a hard case in the *Stack* scenario with 2 boxes. The agent aims to stack the blue box towards the blue spot which is two-stories high.

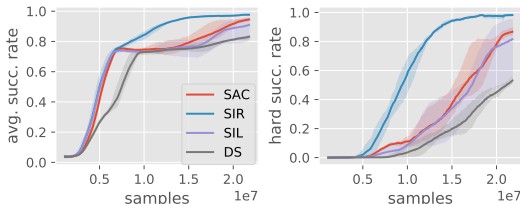

Figure 16: Training results in the *Stack* scenario with a maximum of 2 boxes. We evaluate success rate across training curriculum (left) and in hard cases, i.e., stacking 2 boxes (right).

on a fixed set of 500 *hard* tasks in each scenario and report the average length of successful episodes in Table 8. In *Push* and *Maze* scenarios, these hard tasks are directly sampled from corresponding hard case configurations, while in *Stack* scenario, we only sample the most difficult subset of hard case configurations, i.e. stacking from scratch. SIR achieves comparable mean successful episode length with other algorithms in *Push* and *Maze* scenarios, and is significantly more effective in *Stack* scenario where all other methods fail or struggle to solve.

|  | Pure RL | SIL | DS | SIR |
|---|---|---|---|---|
| Push Unif. | 23.67 | 21.88 | 22.78 | 22.73 |
| Push Mixed 30% | 20.95 | 20.64 | 20.23 | 20.16 |
| Stack (2 boxes from scratch) | 24.21 | 27.00 | 30.54 | 19.16 |
| Stack (3 boxes from scratch) | N/A | N/A | N/A | 39.72 |
| 4-Room maze | 57.43 | 54.94 | 60.40 | 58.54 |

Table 8: Mean length of successful episodes of trained policy evaluated at the same training iterations.

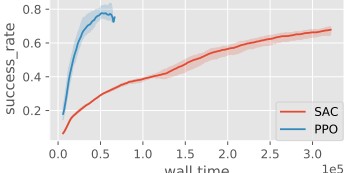

Figure 17: Comparison between SAC and PPO in terms of wall-clock time in *3-Room maze* scenario. The x-axis is measured in seconds.

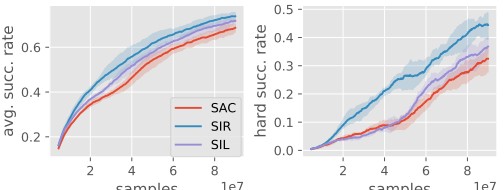

Figure 18: Results of SAC-based algorithms in *3-Room maze*.

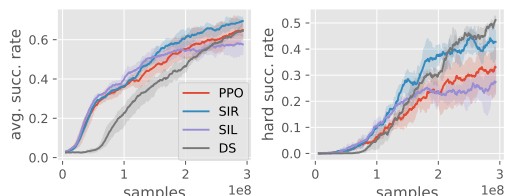

Figure 19: Complete training results of PPO-based algorithms in *4-Room maze*.

