# OpenReview forum: "Solving Compositional Reinforcement Learning Problems via Task Reduction"
_ICLR.cc/2021/Conference — ICLR 2021 Poster_

### Official Review · AnonReviewer4 · 2020-10-28
**Review: Solving Compositional RL problems via task reduction: Some key omissions**

**Rating:** 3
**Confidence:** 3

**Review:**

#######################################################################

Summary:

In this paper the authors propose a method for solving compositional tasks in the RL setting. The method (Self-Imitation via Reduction), is a 2-step method in which the agent first reduces the target task into two simpler tasks, and then solves the full task using as a demonstration the composite task uncovered in step-1.

#######################################################################

Reasons for score:

I am currently voting for a rejection based on what seem to be two key omissions from the paper:

1. The way in which the set of 'library' tasks is selected, and
2. A measure of how expensive this offline component of the algorithm is

The experimental comparisons could also have been stronger.

#######################################################################
Pros:

1. I think the paper is reasonable well written. I found just a few typos, and I think the key conceptual ideas are reasonably easy to follow.

2. The 2-step implementation of some sort of "policy reduction" followed by an imitation learning step would seem to be a good in principle idea that merits further exploration

#######################################################################

Cons:

1. In Sec 3 the authors talk explicitly about a multi-task RL learning setting, and indeed a central part of the method is a search step over interim states $s_\beta$ , but I do not see explicitly how this set is constructed. This is of course a critical consideration for a number of reasons:

(1)  If this set of tasks is large, then you are likely to find a good $s_\beta$, but the search space increases (as does the memory footprint)

(2) If this set is small (and perhaps curated), then the subsequent result is weak (if the agent need only search over a small handful of interim tasks which already include moving the elongated box say, then of course the subsequent learning is rapid).

2. The experimental results would be more compelling if comparisons were made to methods that similarly had some access to interim policies. The comparisons to SAC for example are valid in that they provide a lower bound, but there is additional information available to SIR. Similar it is unclear what the comparison to SAC with a dense rewards shows exactly.

3. The authors mention that the method is extensible even though "... tasks reduction only performs 1-step planning... SIR still retains the capability of learning an arbitrarily complex policy by alternating between imitation and reduction: as more tasks are solved, these learned tasks can recursive further serve as new reductions..." - this is a nice idea, but I saw no evidence of this implementation in the paper.

#######################################################################

Questions during rebuttal period:

Please address the concerns above. Also, you have some comparison with hierarchical policy algorithms in 9.a/b - is it possible to extend these to the other domains?

#######################################################################

Some general comments:

- I think the results for these multi-step methods are often easier to digest and understand if the phases are presented separately:
    - Phase 1, reduction:
        - for each experiment, which task reduction was uncovered by the agent?
        - as you know, in many cases there are multiple valid reductions, which does your algorithm find and why?
        - since your reduction phase is 1-step and greedy, in which situations might it not work so well
        - etc.
    - Phase 2, imitation:
        - comparison to other methods
        - effect of parameter choices
        - etc.
- A little more justification for you particular experimental comparisons can be helpful. It seems like there are a few potential avenues you might have liked to explore:
    - comparison to an information impoverished baseline (like SAC)
    - comparison to different task distributions
    - comparison to other subtask methods
    - etc.

---

> ### Author Response · Authors · 2020-11-13
> **Some important properties of our algorithm might be misunderstood (1)**
>
> Dear reviewer,
>
> We appreciate your comments and suggestions.
>
> In order to better answer your questions, we first explain in detail how our method works in a concrete setting, the 4-Room maze (Fig.6 & Fig.7 in our paper), which we believe can clarify many of your concerns. We then provide detailed answers to your specific questions in the following thread.
>
> **How SIR works in 4-Room Maze:**
>
> 1. We have a *continuous* distribution of tasks. Each task is specified by an agent's initial location, box locations (there are 3 elongated boxes and 1 cubic box),  a target object id (i.e., which object to move), and the goal location. All the locations are specified by 2 coordinates, so a task is parameterized by 12 real parameters and an integer parameter (object id).
>
> 2. For each training episode, a *random* task is selected. If the agent fails to complete the task, we search for an intermediate goal $s_b$, which is specified by the object id (which object to move) and the target position for the object (2 coordinates). *Offline* queries to the learned value function are performed to select the best sub-goal (Eq.1).
>
> 3. We collect successful composite trajectories and run imitation learning on them in addition to RL training. Hence, when the agent learns to manipulate different objects, task reduction helps the agent quickly learn a composite strategy of first moving an elongated box to clear a blocked door and then moving the cubic box. Once the agent learns to move an object across a blocked door, task reduction further helps the agent quickly learn to clear two elongated boxes to clear two blocked doors. Subsequently, the agent gradually learns to manipulate a growing number of objects to accomplish a task.
>
> 4. Finally, the agent learns a complex strategy for an extremely challenging task (Fig. 6), where the agent clears all the 3 elongated boxes before it moves the cubic box to the goal.

---

> > ### Author Response · Authors · 2020-11-13
> > **Some important properties of our algorithm might be misunderstood (2)**
> >
> > ### Responses to specific questions ###
> >
> > **Regarding “the library of tasks”**
> >
> > The task space is continuously-parameterized (i.e., there are multiple continuous parameters). This setting is different from conventional multi-task learning, which typically assumes a fixed, discrete set of tasks. By contrast, our setting is typically considered as goal-conditioned learning [1][2]. Due to this continuous parameterization, our task space contains an infinite number of possible configurations and therefore it is non-trivial to discover the optimal subgoal. To tackle this challenge, we conduct two solutions in this paper: (1) directly search over the entire space with standard planning techniques such as CEM, as what we did in the extension to visual domain and in comparison with HRL on the relatively simple maze with only the agent itself (Sec. 5.3). Note that, the planning problems we consider (i.e., 1-step planning) are even *much simplified* comparing with standard motion planning or HRL literature; (2) we propose an effective and *generic* heuristic for scenarios with an object-centric state representation, i.e., to search for the states with only one object different from the original state (e.g., what we did in the 4-room maze). Note that even with this heuristic the search space remains *huge* since we still need to search over the entire location space of that object.
> >
> > **Regarding “a measure of how expensive this offline component is”**
> >
> > The offline component in our algorithm is just querying a learned value function when evaluating the sub-goal candidates. The cost of this reduction search is not a bottleneck in our method, since all the queries can be fully paralleled as a single input batch to the value network.  We profiled the computing cost w.r.t. wall-clock time in the “Push” scenario and find that reduction search only accounts for approximately 1% of the whole algorithm.
> >
> >
> > **Regarding comparison with methods that also use “interim policies”**
> >
> > We do conduct experiments with baselines having fair access to interim policies.
> > 1. In Sec 5.3 with HRL. The higher-level policy in HRL methods does have direct access to ''interim policies'' by treating ''interim policies'' as low-level policies and *explicitly* proposes sub-tasks to them. While our method *implicitly* learns multiple steps of composition with repeated task reduction and self imitation, which is much lightweight and outperforms HRL baselines.
> > 2. We also consider domains where the interim policies *do not exist at all* for all the methods such as “U-Shape maze” in Fig. 9a. This maze contains no other movable objects except the agent itself. All the algorithms work equally on the observation space that only contains the agent state. So the performances solely depend on how the planning quality of each RL algorithm over the space.
> >
> > **Regarding DS baseline**
> > The comparison to SAC + DS aims to show these problems are challenging, i.e., simple reward engineering might not be sufficient to solve the problem.
> >
> > **Regarding “no evidence of implementation” of recursive reduction**
> >
> > ''Recursion'' *implicitly* occurs when task reduction is performed. Fig 6. shows that our agent masters a complex strategy that interacts with multiple objects to solve a task. Learning such a policy requires multiple task reductions, as we described in the 4-Room maze example. It is a major benefit of our method, i.e., it can implicitly learn increasingly more complex policies via self-imitation without the need of explicitly running multiple task reductions in a single trajectory. Since task reduction only succeeds when both subtasks are solved, the agent must have learned to tackle tasks with (N-1) objects before it learns to handle N-object cases. The recursion happens naturally as training proceeds when previously learned policies can be executed to form part(s) of the composite trajectories in task reduction, thus leading to more complex compositional behaviors.
> >
> > **Regarding “extend hierarchical policies to other domains”**
> >
> > We directly follow their original setting (Fig. 9a/b) for a fair comparison. We did try to run skill chaining in our “Push” scenario, but it struggled to make progress. We also noticed that most HRL works are validated in maze navigation domains and focused on navigating the agent itself to the target position without manipulating other objects. We remark that manipulating multiple objects is very challenging: to our knowledge, there is no other effective deep RL method that solves the sparse-reward stacking problem.
> >
> >
> > [1] Leslie Pack Kaelbling.  Hierarchical learning in stochastic domains:  Preliminary results.  In Proceedings of the tenth international conference on machine learning, volume 951, pp. 167–173, 1993.
> >
> > [2] Tom Schaul, Daniel Horgan, Karol Gregor, and David Silver. Universal value function approximators. In International conference on machine learning, pp. 1312–1320, 2015.

---

### Official Review · AnonReviewer3 · 2020-10-28
**Relevant method, good performance, very limited baselines, and some question remain open.**

**Rating:** 5
**Confidence:** 4

**Review:**

The submission proposes an intuitive curriculum learning method which focuses on sparse reward tasks in RL and uses universal value function approximators.
It has 3 explicit steps:
1. identifying a state to decompose the one task into two
2. solve these new tasks
3. solve the complete task by imitating the trajectories from both subtasks.

On the positive side, the paper is overall clearly written and easy to follow for most parts and performs commensurate or better to baselines on a set of simulated manipulation and locomotion domains.

On the negative side, the method often only performs commensurate or close to the baselines while introducing significant added complexity and additional hyperparameters. The stacking task, which demonstrates the strongest benefit for SIR, requires strong domain knowledge as the search for intermediate states is highly constrained.  Constraining the search for intermediate states to positions with blocks under the required stacking position is significant domain knowledge unavailable to the other methods. The minimum requirement for a fairer comparison would be to include a version of SIR without this constraint.

More generally, aspects regarding the specifics of the space in which we search for intermediate states and the baselines remain unclear (e.g. in terms of the search space since according to the appendix the space for states and goals is not the same and e.g. for stacking other constraints exist).

The final problem regarding the evaluation is that while presenting essentially a curriculum learning method, the paper does not compare against other work in curriculum learning as baseline (e.g. [1,2]).

Other questions remain such as the surprising statement that off-policy SAC underperforms on-policy PPO on the navigation task. Statements that are counter to intuition and existing comparisons between SAC and PPO should be supported with experimental results.

Overall, the introduced method follows a valuable direction for curriculum learning in RL but the submission demonstrates significant weaknesses regarding fair evaluation.

[1] Florensa, Carlos, et al. Automatic goal generation  for reinforcement learning agents. In International Conference on Machine Learning 2018
[2] Racaniere, Sebastien et al. Automated curriculum generation through setter-solver interactions. In International Conference on Learning Representations 2020.


(Disclaimer: I have reviewed a previously submitted version of this work and a big share of critical points remains the same between both reviews including domain knowledge unavailable to baselines and comparison to other curriculum learning methods.)

---

> ### Author Response · Authors · 2020-11-13
> **We did not propose a curriculum learning algorithm**
>
> Dear reviewer,
>
> We really appreciate your comments. We want to clarify that our paper has been substantially updated since NeurIPS. Our method is NOT a curriculum learning method and the term curriculum learning is not even mentioned before Sec. 4. Our method purely tackles compositional RL problems. We appreciate your NeurIPS review and did agree that the previous version made some inaccurate and confusing claims. We believe the current version describes our contributions much more precisely.
>
> Regarding the space for searching intermediate states during task reduction, we want to clarify that we did not use any heuristics in our visual RL task and the navigation without boxes task compared with HRL methods (Sec. 5.3 Fig. 8 & Fig. 9a). In the visual RL task, we follow the standard setting [1] to first learn a VAE latent space and then use the standard planning method CEM to search/plan over the entire latent space. In the navigation experiment without boxes, we similarly search over the entire observation space. In both cases, our method outperforms baseline methods.
>
> For RL problems with a lot of objects and an object-centric state representation, we propose a generic heuristic for efficient search (see Sec. 4.4 “Reduction-Target Search), that is to search for the states with one object different from the current state. In object-centric domains, we find an effective but still general heuristic to reduce the search space,
>
> In “Stack” scenario, the only additional heuristic we use is to exclude the objects already stacked from the reduction target search space. Note that this is just object-level pruning and we still need to search over the entire location space of each remaining object for a good reduction target. In fact, this pruning is not necessary: we include the new stacking results without pruning during reduction target search in Fig. 15 appendix C.  Our method still performs the best and is only slightly less sample-efficient than the original one with pruning. Lastly, we want to emphasize that even the one-object-difference heuristic is not a strong inductive bias empirically: we noticed that the agent even discovered a strategy to manipulate two objects together as demonstrated in the learned behavior (Fig. 4).
>
> Regarding curriculum learning baselines, as we have explained that this work is simply an RL algorithm for solving compositional tasks. Therefore we think it is not necessary to compare with curriculum learning algorithms.
>
> Regarding “off-policy SAC underperforms on-policy PPO on the navigation task”, this is based on our empirical finding that SAC requires a large number of samples to train is a magnitude slower than PPO w.r.t. wall time: SAC requires ~3e5 seconds to achieve success rate 0.6 while PPO trained for 6e4 seconds can already beat it (see Figure 16  in appendix).  We also want to clarify that we did conduct SAC-based experiments in navigation scenarios. We compared SAC-based SIR with other HRL baselines in relatively simple maze navigation domains in the “comparison with HRL” subsection where our method significantly outperforms HRL baselines. For the more complex Room maze, we also add the result of SAC-based experiments (Figure 17) into the appendix, where SIR still performs the best compared with SAC and SAC+SIL.
>
> [1] Soroush Nasiriany, Vitchyr Pong, Steven Lin, and Sergey Levine. Planning with goal-conditioned policies. In Advances in Neural Information Processing Systems, pp. 14814–14825, 2019.

---

> > ### Comment · AnonReviewer3 · 2020-11-24
> > **Feedback**
> >
> > Thank you for the rebuttal.
> > It is appreciated that sometimes different researchers will have different views on what a method represents.
> > To clarify, you propose a method that searches for a good task decomposition such that an intermediate task can be solved to simplify solving the final task. Even if the terminology is not used, this has a lot in common with curriculum learning (i.e. finding easier tasks to simplify learning harder tasks). And because of this similarity, a comparison with methods which include steps for finding simpler tasks (see e.g. the previous reference [1]), such as your algorithm does, would be natural. Instead this type of comparison is missing in the submission. While including any additional algorithm clearly constitutes a good share of additional work, here it is justified given limited baselines baselines.
> >
> > In addition, the strongest benefits are obtained in the stacking task where highly important, additional information is only provided to the proposed method. Here, I appreciate the provision of the additional baseline with reduced privileged information. My suggestion would be to only include the additional curve (also in the main paper) and remove the results with privileged information as these mostly hold information about the quality of the additional information and not the proposed algorithm.
> >
> > The outperforming of SAC by PPO still remains a big question and I expect that this could be addressed with sufficient hyperparameter tuning for the off-policy algorithm, but regarding the paper's focus this aspect can be seen as less important.
> >
> > Overall, this remains an interesting paper with a flawed experimental section, which however has been slightly improved by the additional baseline.

---

> > > ### Author Response · Authors · 2020-11-25
> > > **We have further updated our paper**
> > >
> > > Thanks for your valuable comments.
> > >
> > > **Regarding curriculum learning baseline**
> > > 1. We have updated our paper to include a discussion with the curriculum learning method in Sec 2. We emphasize that despite the conceptual similarity, our method focuses on solving a given task while curriculum learning methods focus on task generation and typically treat policy learning as a black box.
> > > 2. We included a comparison with GoalGAN (reference [1] as mentioned) at the end of Sec 5.3 (Fig 10). Since GoalGAN needs to construct a GAN to generate training tasks, it takes a large number of samples for GAN training. By contrast, our method does not explicitly generate a training task curriculum and therefore learns much faster (even faster than a manually designed training curriculum).
> > >
> > > **Regarding the heuristics in Stacking**
> > > We have updated our paper and now the stacking experiment *does not use ANY* specialized heuristics. The performance almost remains the same.
> > >
> > > **Regarding SAC v.s. PPO**
> > > 1. We add an explanation of this at the beginning of Sec 5.
> > > 2. We want to clarify that we use PPO since it is much *faster w.r.t. wall-clock time* (Fig 17 in Appendix C). Although SAC is typically more sample-efficient than PPO, it runs much more policy optimization steps (i.e., a policy update every a few samples collected) than PPO (i.e., only 1 policy update after a large batch of samples collected). In 4-Room Maze, SAC does not converge after 3 days of training.
> > > 3. We also want to emphasize that we have presented SAC results in a simplified 3-Room Maze in Fig 18 Appendix C. We also run SAC in the comparison with HRL methods.

---

### Official Review · AnonReviewer2 · 2020-10-28

**Rating:** 6
**Confidence:** 4

**Review:**

# Summary
This paper proposes a new method that combines task reduction and self-imitation learning for goal-based reinforcement learning problems. The idea is to decompose a hard task into two subtasks (subgoals) such that the solution to one of them is already known. Self-imitation learning is used to quickly learn to reproduce such successful trajectories. The experimental result shows that the proposed method outperforms the baseline SAC + HER and SAC + SIL as well as hierarchical architectures such as HIRO and DSC.

## Pros
* The idea is interesting and novel.
* The empirical results are good.

## Cons
* Limited to (a subset of) goal-based RL problems.

# Novelty
The proposed idea of decomposing a task into two easy tasks is novel and interesting. It would be worth citing and discussing a relevant prior work [1], which also proposes such a decomposition for goal-based RL.

# Quality
* The empirical results are good. Specifically, it is interesting that the proposed method outperforms hierarchical RL methods without being explicitly hierarchical.
* At the same time, the proposed method seems very specific to a subset of goal-based RL problems, where searching the goal space is computationally tractable.
* Though I appreciate the extension to visual domains using $\beta$-VAE to remedy the aforementioned limitation, the U-Wall maze doesn’t seem like visually complex, and the result (Figure 8) is not very strong. Either showing much better results on Figure 8 or showing results on complex visual domains would strengthen the claim.
* It would be more convincing and interesting to show that the proposed method can do deeper planning by applying task reduction recursively.

# Clarity
* The paper is easy to follow, and the figures are well-presented.
* What is the rationale behind $V(s_a, s_b, g_a) = V(s_a, s_b) * V(s_b, g_a)$? This seems quite specific to “goal-reaching” 1-or-0 reward structures. Do you have an idea how to generalize this to more general reward structures?
* In Algorithm 1, do you generate a new trajectory to get $\tau’$? If this is the case, is this taken into account as the number of steps (x-axis) in the learning curves? Otherwise, they are not fair comparisons.
* Just to check if they are apples-to-apples comparisons, do you use HER across all methods (yours and baselines)?
It would be good to mention that this paper considers goal-based RL problems early in the paper (in abstract or introduction).

[1] Floyd-Warshall Reinforcement Learning: Learning from Past Experiences to Reach New Goals, Vikas Dhiman et al.

---

> ### Author Response · Authors · 2020-11-13
> **Citation added and discussed**
>
> Thank you for pointing out this paper. We have cited it in the related work. This paper considers tabular RL cases and transfers previous experiences to new goals by exploiting a triangularity constraint on value functions. The concept of reusing previous knowledge is similar to ours, but we focus on a different problem that tackles compositional problems (in continuous control).
>
> Regarding the limitation on computationally tractable search spaces, we have applied our method to the visual domain where task reduction is performed over the entire latent space with CEM.
>
> Regarding “it would be more convincing and interesting to show that the proposed method can do deeper planning by applying task reduction recursively”, only requiring a 1-step planning operator for solving complex compositional tasks is an important advantage of our method and makes our method easy to work in practice. We also believe this is one of the reasons why our method outperforms HRL methods, since HRL requires multiple-step planning at a time, which is a substantially harder planning problem. With the help of imitation learning, our policy can still gradually learn increasingly more complex strategies which demonstrate strong compositionality,  for example learning to manipulate multiple boxes in the Stack scenario (Sec. 5.2 Fig. 4) and sequentially pushing away all the elongated boxes that block the door in 4-Room maze scenario (Sec. 5.3 Fig. 6).
>
> “Rationale behind product of values”: in binary sparse reward setting, value function approximates the probability of successfully reaching the goal. Product of two values gives the approximate probability that both reduction trajectories can succeed. In the context of goal-conditioned RL, we believe the binary sparse reward is one of the most general forms of reward functions that does not require any engineering.
>
> Some points to clarify: As we state in the last paragraph of Sec. 5.1, all the rollout timesteps during task reduction are already included in the x-axis of our plots to make a fair comparison. HER is applied in all the SAC-based algorithms (SIR, SAC, SIL).

---

### Official Review · AnonReviewer1 · 2020-10-29
**Task reduction and self imitiation; simple approach with nice empirical results**

**Rating:** 7
**Confidence:** 3

**Review:**

This paper presents Self Imitation via Reduction (SIR) an approach to learning long-horizon tasks by successively reducing it to easy to solve tasks, generating solutions to these easier tasks and self-imitation on successful task solutions. This is done by training a goal-conditioned policy together with a universal value function. When given a task that cannot be solved via the current policy, SIR searches for an intermediate state that divides the task into two easily solvable sub-tasks; this search is carried out by maximising the (composed) value of the sub-trajectories under the current policy. The policy is then executed for these sub-tasks in sequence; success in both these sub-tasks means a solution for the full task is now found. This solution is used as a demonstration that the policy can use for self-imitation via an advantage weighted behavioural cloning objective. This is combined with the policy loss of standard actor-critic algorithms such as SAC and PPO in both the off-policy and on-policy settings respectively and shows significant performance improvements compared to baselines on several long-horizon tasks such as robotic pushing, stacking 3 blocks and a multi-room maze task.

A few points:
1. This approach tackles an important problem of effectively learning policies for long-horizon tasks taking advantage of compositional structure of sub-problems. It nicely combines several ideas from prior work such as self-imitation and universal value functions to present a method that is conceptual simple but performs well empirically on complex tasks.
2. SIR seems quite related to the two-level architecture in standard hierarchical RL approaches — the planner for task reduction is the top level and the low level being the policy. There is a few key difference though: SIR uses this structure primarily for learning and over time the knowledge in this bi-level structure is distilled into the low level policy. It would be interesting if this is discussed a bit further in the paper.
3. Unlike traditional sub-goal selection methods which recursively decompose the problem into sub-problems, SIR does a single reduction step to decompose the task into two sub-tasks. This works well under the assumption that the goal-conditioned policy has sufficient representational capacity to capture the variety of tasks in the environment. Is it possible to extend the current approach to a recursive decomposition for harder tasks?
4. The paper is very well written. There is a clear motivation, contributions and a thorough overview of the related work in this area. The discussions are well structured and together with the appendix a lot of detail is provided on the experiments and methods.
5. A crucial component of the proposed approach is the search for possible reductions. In the proposed approach this is highly structured and limited to very few dimensions of the actual observation space (e.g. only considering object translations in the pushing task, only considering moving unstacked blocks in the stacking task). This provides a key advantage to SIR compared to baselines as it significantly reduces the branching factor of search and consequently can lead to many successful reductions early on during training. This somewhat reduces the strength of the proposed results. As an additional baseline, it would be good to see the performance achieved by SIR when search is not structured and allowed to explore all dimensions of the state space. Does this reduce the performance and/or learning speed of SIR?
6. The paper presents initial results on a vision-based task where a VAE representation is used as state. What is the dimensionality of this representation? As mentioned above, this can have a significant impact on the learning performance (while CEM should do better compared to random search it is not clear if this can mitigate the issue by itself).
7. Another key limitation of the proposed approach is the reliance on arbitrary resets which is not feasible in the real world. While this is briefly discussed in the paper it is not clear how this can be mitigated easily. A more detailed discussion would be useful.

Overall, the approach is quite nice and the initial results are encouraging. I would suggest a weak accept.

---

> ### Author Response · Authors · 2020-11-13
> **Thanks for your comments**
>
> We would like to thank reviewer 1 for your valuable comments and acknowledgement on this work.
>
> 3: “Is it possible to extend the current approach to a recursive decomposition for harder tasks?”  Yes, it is possible to perform multi-step planning to select multiple subgoals at a time but it makes the reduction search a substantially harder planning problem. Only requiring a 1-step planning operator for solving complex compositional tasks is an important property of our method and makes our method easy to work in practice. We also believe this is one of the reasons why our method outperforms HRL methods, which requires multiple-step planning. Also, task reduction offers an alternative perspective on solving compositional problems by only performing 1-step planning at a time but incorporating compositionality through time with self-imitation.
>
> 5 and 6: We are working towards this direction that performs task reduction over the full state in visual domains. In the preliminary U-Wall maze experiment, our algorithm plans with CEM over all dimensions of 16-d VAE latent spaces.
>
> 7: “More detailed discussion on how to mitigate arbitrary reset”:  We can track previous solutions to the trajectories and first return to the step following the tracked solution then perform task reduction, similar to the “return-and-explore” trick in [1]. Also, we only use reset when performing task reduction at training time. During task-time task reduction, we simply start planning from the initial state, as what we have done in Sec. 5.1 “Task Reduction as a Planner”.
>
> [1] Adrien Ecoffet, Joost Huizinga, Joel Lehman, Kenneth O. Stanley and Jeff Clune. Go-Explore: a New Approach for Hard-Exploration Problems. CoRR, abs/1901.10995, 2019.

---

### Author Response · Authors · 2020-11-13
**We have further updated our paper with more results**

We have uploaded a new version of our paper, with all differences marked in blue.

The changes are listed as follows:
1. we include discussions with curriculum learning in Sec 2.
2. we clarify that we use PPO for maze navigation because SAC is much slower than PPO w.r.t. wall-clock time. We explain this in Sec. 5 and Fig 17 in Appendix C. We also include additional SAC results in Fig 18 in Appendix C.
3. We updated the results of the Stacking experiment. So now the task reduction search for stacking does not involve any pruning. The performance of SIR remains unchanged.
4. We include additional results with curriculum learning methods in Sec 5.3 (Fig. 10).

---

### Decision · Program_Chairs · 2021-01-07
**Final Decision**

**Decision:**

Accept (Poster)

**Comment:**

The paper proposes a method for solving challenging sparse reward problems by performing task reduction followed by self-imitation learning from solution trajectories to the reduced tasks.  The core innovation seems to me to be the uses of the reduction search, which is essentially a form of recursive subgoal selection, but where the subgoals are sure to be achievable as assessed by leveraging the learned value function.  This idea seems rather general, though its use is strongly facilitated in this paper by definition of the space (i.e. that object target is the space, is pre-specified; there is only one, rather limited result on a pixel-based task).

Note: Another submission to this conference also explores a quite similar idea to the task reduction proposed in this paper -- see "Divide-and-Conquer Monte Carlo Tree Search".  The breaking down of the problem into sub-problems using the value function is similar, but the details of how the papers proceed from there is quite distinct.

This is a difficult meta-review decision due to the fairly mixed reviews, coupled with limited engagement in the discussion phase.  Two reviewers felt the paper was solid and could be accepted (R1 and R2 with scores 7 and 6 respectively).  R3 gave a borderline review that leaned towards reject (score 5).  R3 replied to the initial author response, which provided helpful feedback to the authors. Ultimately, in my assessment, the authors did a fairly thorough job of addressing some of the points raised by R3, including by adding an additional comparison even where they didn't agree with the reviewer. R4 assigned the paper the lowest score of 3.  The authors provided a lengthy reply to this review asserting that the review may have reflected misunderstanding of paper details, but the reviewer did not respond to the authors.

Two core issues raised about this paper relate to the definition of the space for subgoals and the limited difficulty of the tasks. However, this method does not claim to be entirely ignorant of the task space so I don't see the fact that they do include some domain knowledge in designing the goal space to be totally undermining of the method.  They focus on the complementary issue of how to break down difficult problems into sub-problems.  While it would be considerably more impressive if the goal space were learned, I think this harder version of the problem remains a fundamental and deep problem within AI, so it seems to me too much to ask of the present paper (especially given that it was not the stated focus of the paper).  And while the tasks explored in the paper are a little contrived (some repetitive motifs and designed with a relatively small search space over subtasks), these problems do have some complex structure.  Compared to many works in this field, I applaud the authors for engaging with problems with both long-horizon task structure as well as complex high-DoF continuous control component.

While I agree with some of the concerns raised, my overall assessment is that I find the contributions sufficiently innovative and substantial to justify acceptance.  The authors proposed a specific innovation and evaluated that innovation.  Insofar as their innovation is somewhat general, I don't think this paper can be the last word on how well it compares with the diverse approaches it could be set against.  And while the experiments are not definitive, I do think they do constitute a fairly ambitious initial validation of the core idea.